# Failure Mechanism of pHEMT in Navigation LNA under UWB EMP

**DOI:** 10.3390/mi13122179

**Published:** 2022-12-08

**Authors:** Yonglong Li, Bingrui Yu, Shengxian Chen, Ming Hu, Xiangwei Zhu, Xuelin Yuan

**Affiliations:** School of Electronics and Communication Engineering, Sun Yat-Sen University, Shenzhen 518107, China

**Keywords:** UWB EMP, GaAs pHEMT, LNA, gain compression

## Abstract

With the development of microelectronic technology, the integration of electronic systems is increasing continuously. Electronic systems are becoming more and more sensitive to external electromagnetic environments. Therefore, to improve the robustness of radio frequency (RF) microwave circuits, it is crucial to study the reliability of semiconductor devices. In this paper, the temporary failure mechanism of a gallium arsenide (GaAs) pseudomorphic high electron mobility transistor (pHEMT) in a navigation low-noise amplifier (LNA) under the jamming of ultra-wideband (UWB) electromagnetic pulses (EMP) is investigated. The failure process and failure mechanism of pHEMT under UWB EMP are elaborated by analyzing the internal electric field, current density, and temperature distribution. In detail, as the amplitude of UWB EMP increases, the output current, carrier mobility, and transconductance of pHEMT decrease, eventually resulting in gain compression. The injection experiment on LNA, which effectively verified the failure mechanism, indicates that the gain of pHEMT is suppressed instantaneously under the jamming of UWB EMP and the navigation signal cannot be effectively amplified. When UWB EMP amplitude increases to nearly 10 V, the BeiDou Navigation Satellite System (BDS) carrier signal is suppressed by nearly 600 ns. Experimental results accord well with the simulation of our model. UWB EMP jamming is a new and efficient type of electromagnetic attack system based on the device saturation effect. The performance degradation and failure mechanism analysis contribute to RF reinforcement design.

## 1. Introduction

In the new global electromagnetic environment, intentional electromagnetic interference (IEMI) has become a central issue for electronic systems. A strong electromagnetic pulse can be coupled into electronic systems through antennae, sensors, and gaps in the package shell of the system. Ultra-wideband electromagnetic pulse (UWB EMP) [1,2,3], high-altitude electromagnetic pulse (HEMP) [4,5,6], or high-power microwave (HPM) [7,8,9,10,11] can generate a strong directional electromagnetic pulse to cause electronic systems to generate high voltage or large current in the circuit, resulting in system failure or unrecoverable thermal damage.

Whether from a military or civilian perspective, the compatibility and reliability of electronic equipment and electronic systems in the EMP environment are of great concern. The fundamental cause of the failure or burnout of the system is damage to semiconductor devices in the circuit. Therefore, investigation of failure mechanisms for devices is the basic premise to ensure the normal operation of a receiving system. In recent years, a growing number of scholars have studied the jamming of electromagnetic pulses on semiconductor devices through simulation and experiment. Li et al. studied the damage effect characteristics of GaAs pHEMT under C-band high-power microwave. It is concluded that the gate metal in the first stage of the device is the most vulnerable to HPM damage [12]. Qin et al. established an enhancement-mode p-gate AlGaN/GaN HEMT to investigate the self-heating and HPM effects from both electrical and thermal aspects. The results showed that the HPM effect would lead to breakdown and irrecoverable thermal damage of the source and drain regions below the gate, while self-heating could only cause thermal accumulation in the drain region [13]. Yu et al. investigated the nonlinear degradation of an L-band GaAs-based LNA caused by EMP. The experimental results demonstrated that the real-time response, RF, and DC (direct current) characteristics of LNA samples under the jamming of EMP showed nonlinear and permanent degradation features [5]. Pan et al. developed an electrothermal coupling model of CMOS inverters and found that the latch-up effect is the main mechanism of failure of the CMOS inverter by studying the temperature and current characteristics [14]. Zheng Quan studied the limiting characteristics and damage mechanism of non-punch-through PIN diodes under HPM-UWB [15]. As for an electric system, LNA is a key element in the RF front end of the wireless receiver, which is generally the most sensitive section at the same time. However, most of the studies on IEMI have focused on the power threshold of burnout of the device, and few of them have been able to draw on any systematic research into the failure mechanism of LNA under UWB EMP. Our primary concern is the suppression effect of UWB EMP on the device gain. Although the pulse width of UWB EMP is in the nanometer scale and the average power is smaller than that of HPM, the wide bandwidth of UWB EMP and a large amount of antenna coupling may also cause instantaneous jamming to the electronic system. When a useful signal is suppressed, the electronic system may be paralyzed. Thus, it is essential to understand the failure mechanism of the gain suppression caused by UWB EMP.

This paper is organized as follows. We first give a brief overview of recent research on electromagnetic pulse interference. Subsequently, models for the simulation are described in Section 2, including the device structure, numerical model, and signal model. In Section 3, to accurately describe the internal microscopic behavior of the HEMT, the electric field strength, current density, and temperature characteristics of the HEMT are performed by Sentaurus-TCAD software (an advanced multidimensional device simulator). The failure mechanism of the device under the jamming of UWB EMP is explained, and the cause for the gain compression of the HEMT device is summarized. Section 4 presents the results of the experiment of UWB EMP injection into LNA, which demonstrates the failure mechanism in introduced Section 3, i.e., after the HEMT is suppressed by the UWB EMP, the gain drops significantly, but the device does not burn out. Finally, the conclusion gives a brief summary and critique of the findings to provide references for the RF front-end protection.

## 2. Device Structure and Simulation Model

As the core component of LNA, HEMT has the superior performance of high electron mobility, high gain, high power, low noise, and low on-resistance. It is playing an extremely important role in navigation, satellite communication, remote sensing, and military electronic countermeasures [16,17,18,19,20]. As the first-stage device of the RF front-end, LNA is vulnerable to the jamming of high-power electromagnetic pulse, making it vital to study its failure mechanism.

Based on semiconductor physics, the transient response of pHEMT under the jamming of UWB EMP can be obtained by numerical simulation using Sentaurus-TCAD software so as to observe the whole process of device failure. The simulation results can provide a reference for failure mechanism analysis, while the distribution of internal physical quantities cannot be attainable by experiments.

### 2.1. Device Structure

In this paper, a typical δ-doping AlGaAs/GaAs pHEMT is established, and the basic structure of the device in Sentaurus-TCAD software is illustrated in Figure 1.

The cross-section reveals the internal of the HEMT: from bottom to top, a 0.8 μm GaAs substrate, a 10 nm InGaAs channel layer, a 34.5 nm AlGaAs spacer layer, and a 30 nm GaAs cap layer. The surface of the HEMT is passivated by 50 nm Si_3_N_4_ and 0.1 μm oxide. To make the heterojunction quantum well deeper to increase the electron concentration in the two-dimensional electron gas (2EDG), a 2 nm δ-doping layer is also added to the AlGaAs spacer layer. In addition, the gate length is 0.25 μm, the Schottky barrier height is 0.9 eV, and the mole fractions of Al_x_Ga_1−x_As and In_1−y_Ga_y_As are 0.30 and 0.75, respectively.

The DC characteristics of the device are analyzed in Figure 2, which demonstrates the accuracy and reliability of the device model.

### 2.2. Numerical Model

To study the failure mechanism of HEMT under UWB EMP, the basic physical equations to be solved include the Poisson equation, continuity equation, transport equation, etc. The physical models to be considered include the mobility model, composite model, carrier generation model, and carrier statistical model. The physical equations and model descriptions of these models can be found in [21,22].

Since lattice self-heating and energy balance should be taken into account when UWB EMP is injected into the devices, the electron and hole current density equations should be modified as:(1)$$\begin{array}{l} \vec{J_{n}}=-nq\mu_{n}\left(\nabla \phi_{n}+P_{n}\nabla T\right)\\ \vec{J_{p}}=-pq\mu_{p}\left(\nabla \phi_{p}+P_{p}\nabla T\right) \end{array}$$
where *μ*_n_ and *μ*_p_ refer to the mobility of electrons and holes; *Φ*_n_ and *Φ*_n_ denote the quasi-Fermi potentials of electrons and holes; *P*_n_ and *P*_p_ are the absolute thermoelectric powers of electrons and holes; and T represents the temperature.

Due to the self-heating effect inside the device, the thermodynamic model should be solved, and the lattice temperature can be calculated by:(2)$$\begin{array}{c} \frac{\partial }{\partial t}c_{L}T-\nabla \cdot \kappa \nabla T=-\nabla \cdot \left[\left(P_{n}T+\phi_{n}\right)\vec{J_{n}}+\left(P_{p}T+\phi_{p}\right)\vec{J_{p}}\right]-\left(E_{C}+\frac{3}{2}kT\right)\nabla \cdot \vec{J_{n}}\\ -\left(E_{V}+\frac{3}{2}kT\right)\nabla \cdot \vec{J_{p}}+qR_{\mathrm{net}}\left(E_{C}-E_{V}+3kT\right) \end{array}$$
where *c*_L_ refers to the lattice thermal capacity; *κ* denotes the thermal conductivity; *k* denotes the Boltzmann constant; *E*_C_ and *E*_V_ are the conduction-band and the valence-band energy levels; and *R*_net_ is the net electron–hole recombination rate.

### 2.3. Signal Model

UWB EMP generally refers to electromagnetic waves with rise time and duration in a nanosecond or sub-nanosecond scale, and the spectrum ranges from tens of megahertz to several gigahertz. It has the characteristics of fast leading edges, narrow pulse width, and wide spectrum [23]. UWB EMP is usually expressed by Gaussian pulse, with simple expressions and rich spectrum components. When the mean value is 0 and the variance is *σ*^2^, the signal can be expressed as:(3)$$f(t)=\frac{1}{\sqrt{2\pi }\sigma }e^{-\frac{t^{2}}{2\sigma^{2}}}=f(t)=\frac{\sqrt{2}}{\alpha }e^{-\frac{2\pi t^{2}}{\alpha^{2}}}$$
where *α* = 4π*σ*^2^ denotes the Gaussian pulse width.

In engineering, the first or second derivative of Gaussian pulse is commonly used to represent UWB EMP [24,25]. Figure 3 illustrates the time-domain and frequency-domain characteristic waveforms of the first derivative of Gaussian pulse, which approximates the waveform of UWB EMP used in this paper.

After the device structure of pHEMT is established, the simulation circuit of UWB EMP effect shown in Figure 4 is built. Initially, the source is grounded, and a bias voltage of 2 V is applied to the drain. A nanosecond UWB EMP is injected from the gate of the pHEMT to simulate the process of coupling the UWB EMP to the input port of the pHEMT LNA through the front-door path. By observing the current, voltage, and temperature distribution inside the device under the jamming of UWB EMP, the changes in device transconductance, carrier mobility, and gain are analyzed.

## 3. Failure Mechanism Analysis

### 3.1. UWB EMP Failure Effect

Based on the simulation model given above, when a train of UWB EMP with an amplitude of 8.5 V, a pulse width of 1 ns, and a repetition frequency of 100 kHz is injected, the variation in peak temperature over time inside the pHEMT is shown in Figure 5.

When the UWB EMP arrives, the peak temperature inside the device rises sharply and drops rapidly after the pulse. The reason for the second rise after the temperature drop is that the device is still in the conduction state with the UWB voltage at 0 V at the time. It can be seen from Figure 5 that the internal temperature of the HEMT only accumulates in the first few cycles. After a period of time, the heat accumulated by the UWB EMP and the relaxation time of the internal cooling of the device reaches a balance. Thus, the heat no longer accumulates, and the internal temperature of the device no longer rises.

Moreover, the pulse width of UWB EMP is only 1 ns, and the energy injected into the device is limited. Therefore, after reaching thermal equilibrium, the internal temperature of the device will not reach the melting point of GaAs material (1511 K), which means the failure degradation of the device is reversible.

To study the failure process of the device, the half cycle of UWB EMP is taken as an example. When the UWB EMP with a voltage amplitude rising from 0 V to 10 V within 1.5 ns is injected into the device, the distribution of the internal electric field and current of the device at different times are shown in Figure 6 and Figure 7, respectively.

It can be seen from the figure that with the increase of the amplitude of the UWB EMP, the electric field in the vertical direction increases gradually. When the amplitude of UWB EMP further increases, breakdown occurs in the AlGaAs/InGaAs heterojunction and Schottky junction, which makes the high-mobility electrons in the 2EDG below the gate move vertically. Due to the limited capability of carrier transport, a large number of electrons generated by breakdown are not transported to the electrode in time, but accumulate below the gate, forming a conductive channel in the vertical direction between the gate and the InGaAs channel.

The heat generated in the transistor can be expressed as Q = J·E. Figure 8 shows the temperature distribution inside the device. As can be seen from the figure, the temperature reaches a peak below the gate and near the source, which is due to the fact that the 2 V bias voltage at the drain makes the gate current pass through the two-dimensional electron gas channel and get closer to the source.

### 3.2. UWB EMP Failure Mechanism Analysis

Since the output current of the drain is a function of the electron concentration and electron velocity of the 2EDG in the InGaAs channel [26], the UWB EMP injected into the gate will not only accumulate heat inside the device but also affect the output current, causing temporary failure of the device, as shown in Figure 9.

When the UWB EMP starts to be injected, the voltage is low, and the Schottky junction is under positive bias, so the output current increases linearly with the increase of the UWB EMP voltage. As the UWB EMP voltage increases further, the drain reduces to low potential, and the output current saturates and even starts to decrease in reverse.

The reason for the above phenomenon is that when the UWB EMP voltage reaches the threshold, the increase of the electric field in the vertical direction and the lattice temperature inside the device increase the scattering of carriers, which leads to the decrease of carrier mobility. The field in the horizontal direction makes the carrier transport velocity saturated, resulting in saturation of the output current of the drain. Combined with Figure 6 and Figure 7, or analysis, the breakdown occurs in the AlGaAs/InGaAs heterojunction and Schottky junction, the gate becomes a high-voltage terminal, and the drain becomes a low-voltage terminal, causing a gradual decrease in output current. In addition, as the carrier mobility lowers, the transconductance *G*_m_ also decreases, leading to a lessening of gate control capability [27], as shown in Figure 10. Thus, the gain of the device decreases due to the positive correlation with transconductance *G*_m_, so the phenomenon of gain compression emerges, as shown in Figure 11.

As shown in Figure 10 and Figure 11, the injection of the UWB EMP leads to the degradation of the electron transport characteristics, and the threshold voltage shifts to the left. The transconductance increases at low UWB EMP amplitude and decreases when the amplitude of the UWB EMP reaches a certain level. Subsequently, the *G*_m_ decreases more significantly, corresponding to the gain compression phenomenon. The gain increases slightly after the decrease because of the abnormal current generated by the electrons from the breakdown in the vertical direction of the gate and the InGaAs channel. Overall, the device gain is kept at a low level by the jamming of the UWB EMP.

## 4. LNA Effect Experiment under UWB EMP

The effect experiment can monitor the output performance of the device in real time, which is a relatively intuitive method to study the law of UWB effect. The effect data obtained by experiments are more realistic and have high confidence. In this section, through injection experiment, the self-developed UWB EMP with a pulse width of fewer than 1 ns is injected into the LNA whose core component is GaAs pHEMT, and it further demonstrates that the gain of HEMT decreases greatly under the jamming of UWB EMP in the simulation in Section 3. The jamming effect of UWB EMP on the BDS-III navigation carrier signal is tested, and the suppression is observed.
(4)$$Gain=20\mathrm{lg}\frac{\left|V_{\mathrm{out}}\right|}{\left|V_{\mathrm{in}}\right|}$$

The gain of the amplifier can be expressed as the ratio of the output voltage to the input voltage, as shown in Equation (4). The failure of LNA can be determined by judging whether there is a significant drop in the gain. After stopping the injection of the pulse, it can be observed whether the signal can be recovered to determine whether the LNA is burned out or not. The waveform of the injected UWB EMP in the experiment is shown in Figure 12.

In the experiment, we manufactured the LNA sample by ourselves using a pHEMT-based low-noise MMIC amplifier (MCL 545G) produced by mini-circuits, which has a unique combination of low noise and high IP3, making this amplifier ideal for the receiving front-end circuit of the satellite positioning system. The operating frequency is 1.1–1.6 GHz, the amplification gain is 30 dB, and the noise figure is 1.0 dB. The photo and circuit schematic of the LNA sample is shown in Figure 13. This design operates on a single +5 V supply (VCC) and is internally matched to 50 Ohms.

The injection experiment is carried out at a room temperature of 27 °C. To prevent the equipment from being damaged by UWB EMP, it is necessary to connect attenuators to the output port of the UWB EMP source. Then, the UWB EMP and BDS-III carrier signal are simultaneously injected into the LNA, as shown in Figure 14.

The experimental results show that when the UWB EMP, with a pulse width of less than 1 ns and amplitude of about 2 V, is injected into the LNA through the attenuator, the carrier signal is distorted and the gain compression begins to appear. When the amplitude increases to nearly 10 V, the BDS-III carrier signal is suppressed for nearly 600 ns, and the LNA resumes the amplification function after a period of time, as shown in Figure 15. The higher the amplitude, the longer the suppression time. The repetition frequency of UWB EMP has little effect on the duration of suppression under the jamming of a single pulse, but for the same pulse amplitude, the larger the repetition frequency, the better the suppression effect of the whole signal.

The above results verify the failure mechanism of UWB EMP on HEMT devices introduced in Section 3. When the amplitude of the UWB EMP is injected into the LNA and reaches a certain value, the navigation signal becomes distorted and the gain is compressed, with the result that the signal cannot be effectively amplified. As the amplitude of the UWB EMP increases, a large number of electrons in the device are captured by the defects and cannot get rid of shackles immediately, so the output current cannot respond to the jump of the UWB EMP, further increasing the duration of suppression. In a period of time after the injection of pulse, the LNA gradually resumes its amplification function. In other words, the LNA has a reversible failure effect.

## 5. Conclusions

With the prosperity and progress of jamming technology, the threat of EMP environment to integrated circuits and electronic systems is becoming more and more significant. In this paper, the failure effect and failure mechanism of pHEMT in navigation LNA under the jamming of UWB EMP are investigated. The variation in the electric field, current, and temperature inside the device is simulated by Sentaurus-TCAD software. The simulation results show that the failure mechanism of UWB EMP is the gain suppression of the device. In detail, when the UWB EMP voltage reaches a certain value, the output current, the carrier mobility, and the *G*_m_ decrease, resulting in the gain compression and DC performance degradation of the device.

Finally, the failure mechanism analysis is demonstrated by injection experiments. In other words, the LNA has a reversible failure effect. The BDS-III carrier signal is suppressed for nearly 600 ns when the UWB EMP amplitude increases to nearly 10 V. The higher the amplitude, the longer the suppression time. In fact, since the external circuit of the device is not pure resistance, the UWB EMP will generate additional harmonic components in the circuit. After the navigation signal passes through the process of mixing, filtering, and amplification, the pulse width of UWB EMP will be further expanded in the electronic system, and the system-consumed power will increase, causing the duration of suppression of LNA to be further increased.

UWB EMP jamming is a new and efficient type of electromagnetic attack system based on the device saturation effect. This is completely different from the mechanism of the jamming of other HPM burnt devices. This paper has provided a deeper insight into the reliability of the pHEMT in navigation LNA under UWB EMP. The research results are helpful in strengthening the protective measures of EMP on the RF front end, especially the suppression effect of UWB EMP on device gain.

## Figures and Tables

**Figure 1 micromachines-13-02179-f001:**
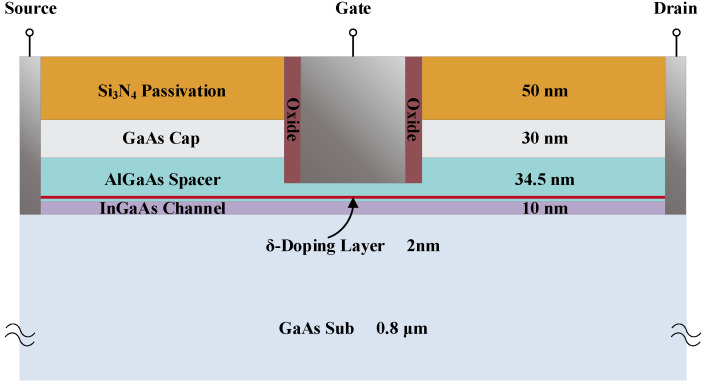
The basic structure of δ-doping AlGaAs/InGaAs pHEMT.

**Figure 2 micromachines-13-02179-f002:**
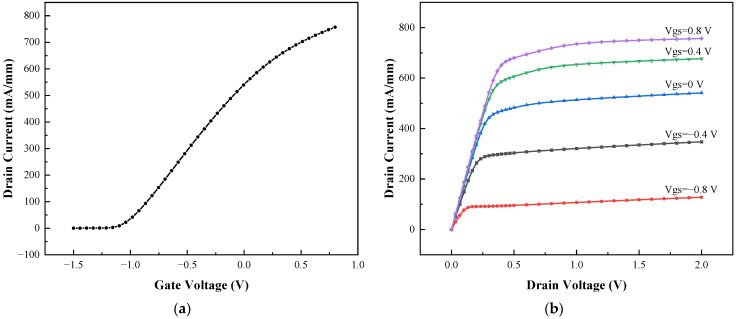
DC characteristic of pHEMT: (**a**) opening characteristic curve; (**b**) transfer characteristic curve.

**Figure 3 micromachines-13-02179-f003:**
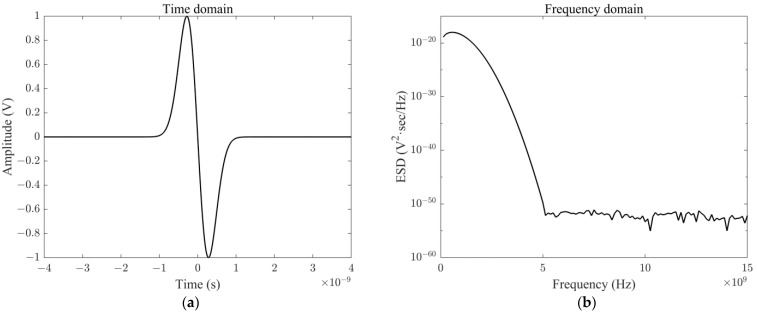
UWB characteristic waveform: (**a**) time domain; (**b**) frequency domain.

**Figure 4 micromachines-13-02179-f004:**
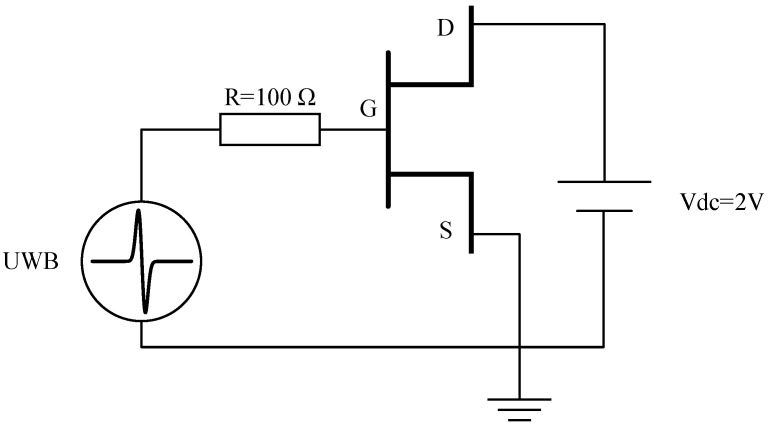
Schematic diagram of simulation circuit.

**Figure 5 micromachines-13-02179-f005:**
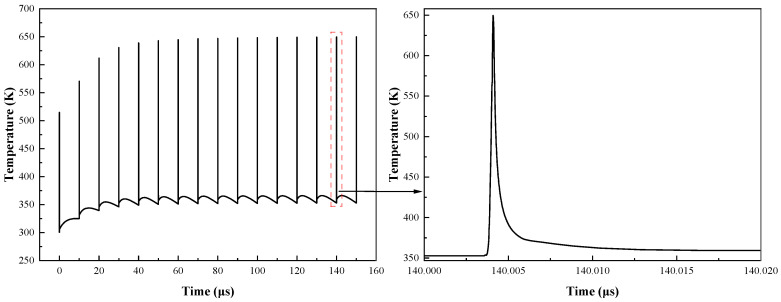
Variations in peak temperature over time inside the pHEMT.

**Figure 6 micromachines-13-02179-f006:**
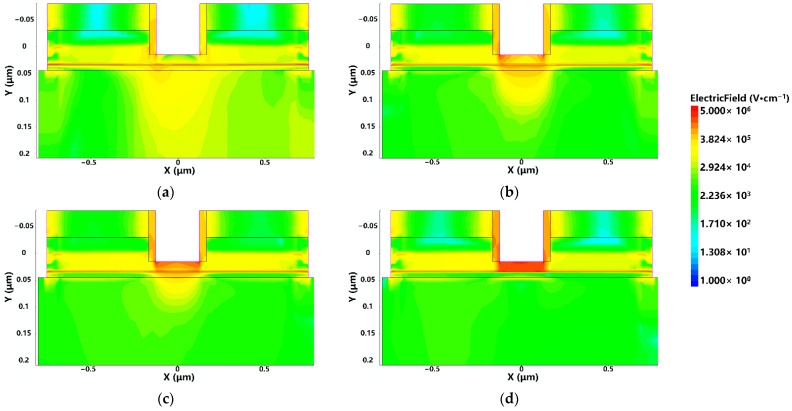
The distributions of electric field strength: (**a**) 0.7 ns; (**b**) 1.0 ns; (**c**) 1.2 ns; (**d**) 1.4 ns.

**Figure 7 micromachines-13-02179-f007:**
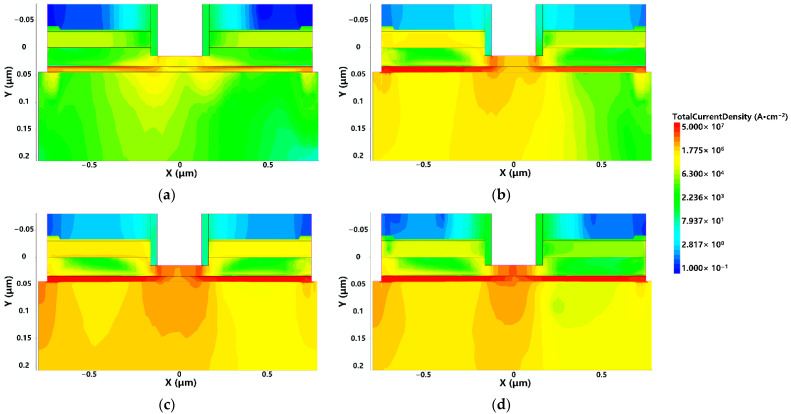
The distributions of current density: (**a**) 0.7 ns; (**b**) 1.0 ns; (**c**) 1.2 ns; (**d**) 1.4 ns.

**Figure 8 micromachines-13-02179-f008:**
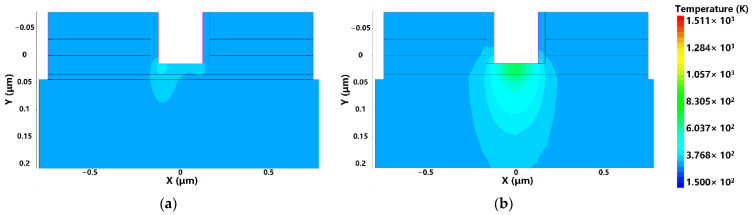
The distributions of temperature: (**a**) 1.0 ns; (**b**) 1.4 ns.

**Figure 9 micromachines-13-02179-f009:**
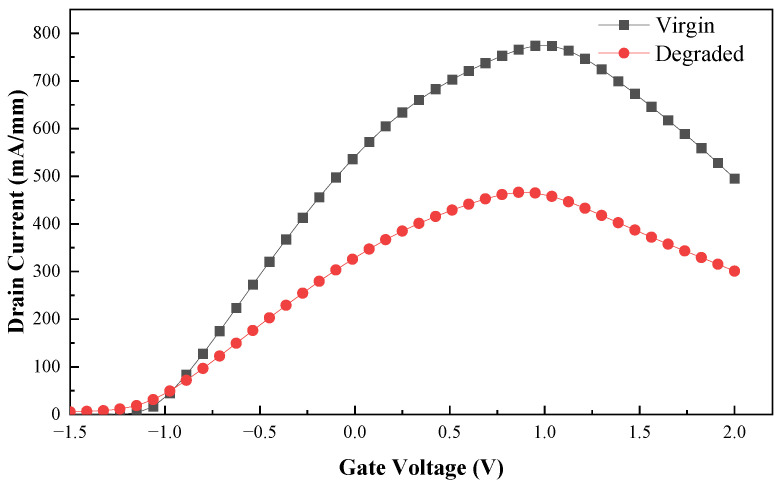
The direct current under UWB pulse jamming.

**Figure 10 micromachines-13-02179-f010:**
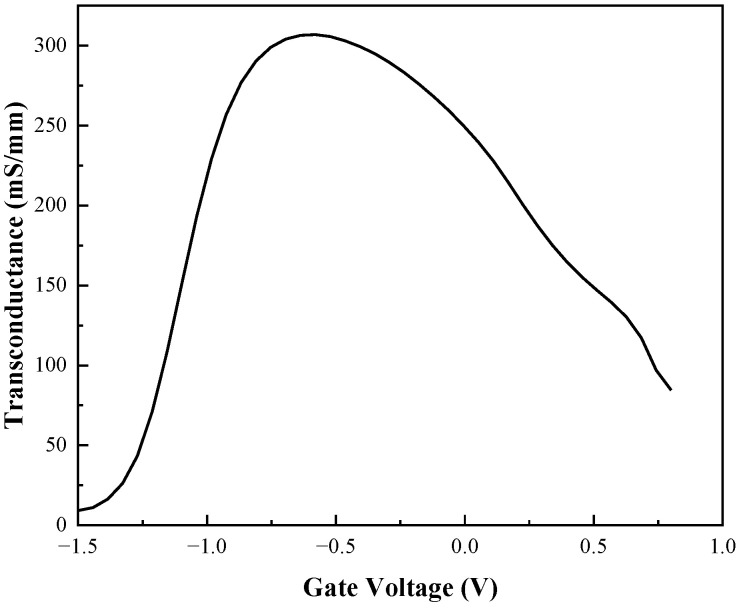
Behavior of *G*_m_ vs *V*_gs_.

**Figure 11 micromachines-13-02179-f011:**
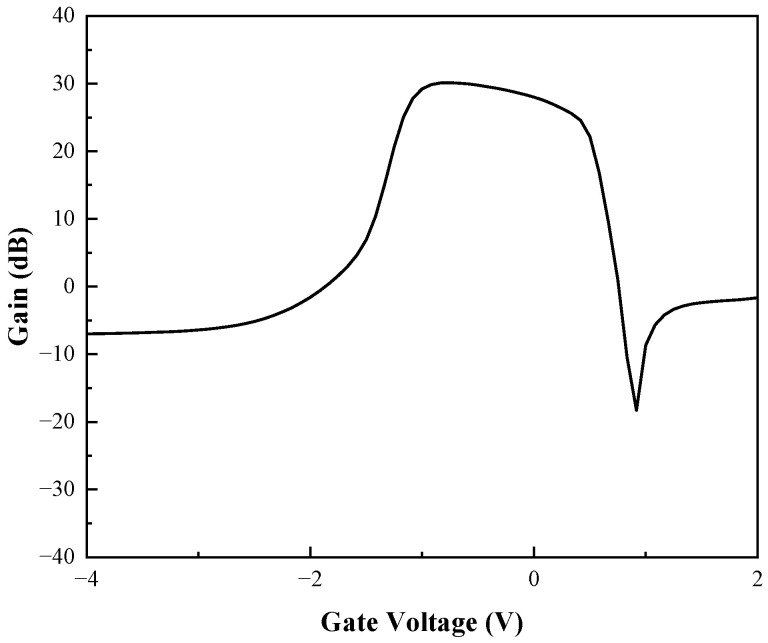
Behavior of Gain vs *V*_gs_.

**Figure 12 micromachines-13-02179-f012:**
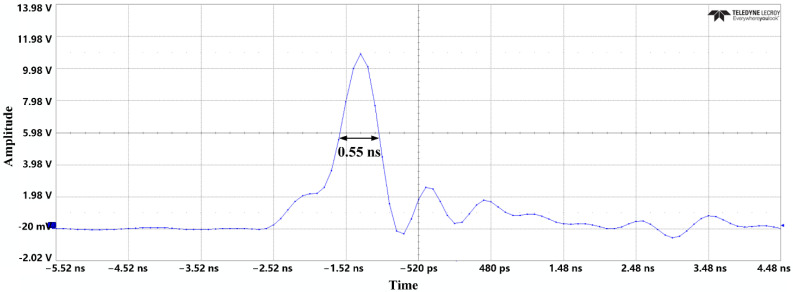
UWB attenuation waveform.

**Figure 13 micromachines-13-02179-f013:**
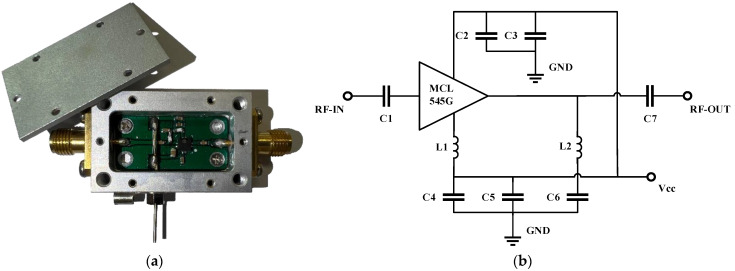
LNA sample: (**a**) photo; (**b**) circuit schematic.

**Figure 14 micromachines-13-02179-f014:**
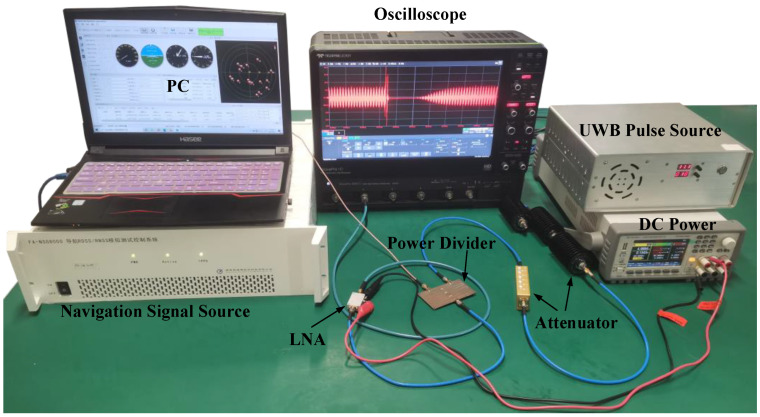
Experiment environment for the LNA.

**Figure 15 micromachines-13-02179-f015:**
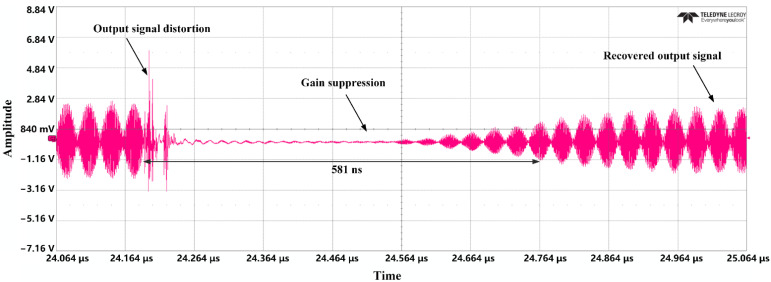
Effect experiment results.

## Data Availability

The data used to support the findings of this study are available from the corresponding author upon request.

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
