# Peer review of "Failure Mechanism of pHEMT in Navigation LNA under UWB EMP"

_micromachines, 2022, doi:10.3390/mi13122179_

Round 1

Reviewer 1 Report

This paper proposed a failure mechanism of pHEMT in navigation LNA under UWB EMP. Although the experiment is complete and thorough, the manuscript lacks novelty. In addition, no comparison table or specification summary is given in this manuscript.

Author Response

Dear editors and reviewers

We would like to thank you for your efforts in reviewing our manuscript titled “Failure Mechanism of pHEMT in Navigation LNA under UWB EMP” and for providing many helpful comments and suggestions, which will all prove invaluable in the revision and improvement of our paper, as well as in guiding our research in the future.

Considering the comments on the manuscript, a more detailed explanation of the numerical simulation methods and effect experiment is introduced in this paper. Besides, all the mistakes mentioned in the comments have been well-corrected. All the revised and supplementary content are marked up using the "Track Changes" function, and highlights are highlighted in yellow. We sincerely hope that the revised manuscript will help you understand our work better.

Appended to this letter is our point-to-point response to the comments raised by the reviewers. Thank you again for your invaluable comments and suggestions. I look forward to hearing from you soon in due course.

Yours sincerely,

Yonglong Li,

liylong27@mail2.sysu.edu.cn,

College of Electronics and Communication Engineering,

Sun Yat-sen University – Shenzhen Campus.

Response to Reviewer1 Comments

Point 1: This paper proposed a failure mechanism of pHEMT in navigation LNA under UWB EMP. Although the experiment is complete and thorough, the manuscript lacks novelty. In addition, no comparison table or specification summary is given in this manuscript.

Response 1: We are grateful to the reviewer for evaluating the manuscript and apologize for not clearly showing the novelty of our work. In the following, we will give a clearer and more concise explanation of our work.

(1) About the focus of research

As for an electric system, LNA is a key element in the RF front-end of the wireless receiver, which is generally the most sensitive section at the same time. However, most of the studies in the IEMI have focused on the power threshold of burnout of the device, and few of them have been able to draw on any systematic research into the failure mechanism of LNA under UWB EMP.

Our primary concern is the suppression effect of UWB EMP on the device gain. Although the pulse width of UWB EMP is in the nanometer scale and the average power is smaller than that of HPM, the wide bandwidth of UWB EMP and a large amount of antenna coupling may also cause instantaneous jamming to the electronic system. When the useful signal is suppressed, the electronic system may be paralyzed. Thus, it is essential to understand the failure mechanism of the gain suppression caused by UWB EMP.

The simulation results show that the failure mechanism of UWB EMP is the gain suppression of the device. In detail, when the UWB EMP voltage reaches a certain value, the output current, the carrier mobility and the Gm decrease, resulting in the gain compression and DC performance degradation of the device.

The failure mechanism analysis is demonstrated by injection experiments; in other words, the LNA has a reversible failure effect. The BDS-â…¢ carrier signal is suppressed for nearly 600 ns when the UWB EMP amplitude increases to nearly 10 V. Experimental results accord well with the simulation of our model.

UWB EMP jamming is a new and efficient type of electromagnetic attack system based on the device saturation effect. This is completely different from the mechanism of the jamming of other HPM burnt devices. This paper has provided a deeper insight into the reliability of the pHEMT in Navigation LNA under UWB EMP. The research results of this paper are helpful is strengthening the protective measures of EMP on the RF front-end, especially the suppression effect of UWB EMP on device gain.

The partial supplementary and highlighted content could be viewed on lines 37-38, lines 58-68 and lines 280-301.

(2) About research methodology

We focus on the failure mechanism of pHEMT in Navigation LNA under UWB EMP by means of Numerical simulation and experimental demonstration.

Numerical simulation: Numerical simulation plays a very important role in the study of the effect mechanism of EMP in microelectronic devices. Based on the semiconductor physics equation, the transient response of pHEMT under the jamming of UWB EMP can be obtained by numerical simulation using Sentaurus-TCAD software so as to observe the whole process of device failure. The simulation results can provide a reference for failure mechanism analysis of the device, while the distribution of internal physical quantities cannot be attainable by experiments.

Experimental demonstration: The effect experiment can monitor the output performance of the device in real-time, which is a relatively intuitive method to study the law of the UWB effect. The effect data obtained by experiments are more realistic and have high confidence.

The partial supplementary content could be viewed on lines 89-93 and lines 227-229.

(3) About comparison table or specification summary

Considering the innovativeness of this analysis when applied to peer review reports, we did not have much previous research on which to rely. So, no comparison table is given in this manuscript. UWB EMP jamming is a new and efficient type of electromagnetic attack system based on the device saturation effect. In particular, there is a lack of theoretical research on the jamming response of high repetition frequency of UWB EMP to the RF front-end. To a certain extent, it limits the wider application and development of the high repetition frequency of UWB EMP.

The partial supplementary and highlighted content could be viewed on lines 58-68.

Reviewer 2 Report

The paper is interesting and includes both theoretical and experimental contributions to investigate the operation of a receiver, in particular the LNA stage which is likely to be the first to be hit by an incoming EM pulse.

My most compelling comment to the study reported in this paper is that the focus on pHEMT is not supported by clear evidence that the hardware used in the experimental verification is effectively based on a pHEMT device. Lines 229-231 describe the LNA (including a picture in Fig. 13) but there is no indication of the manufacturer or source of the LNA, leaving the reader without the opportunity to replicate the results of the paper.

Other less critical comments I would offer to the authors:

Line 128 states that "the first or second derivative of Gaussian pulse is commonly used to represent UWB EMP [24,25]. Figure 3 illustrates the time and frequency domain characteristic waveforms of UWB EMP, which approximates the waveform used in this paper." The authors should specify that Fig. 3 shows the first derivative.

Fig. 5 shows the temperature variation of the device excited by a train of impulses. However, lines 143-154 do not confirm that the excitation is a train of impulse. While the explanation of the authors in those lines is logical, the type of excitation the device is subject to, is not clearly defined: is it a train of pulses? Is my interpretation correct? The authors should clarify.

Eq. (4) should be referred to as voltage gain and it should read 20*log |Vout|/|Vin|.

Line 231 should report the noise figure as 1dB rather than just 1 - or provide the correct interpretation of that number.

Author Response

Dear editors and reviewers

We would like to thank you for your efforts in reviewing our manuscript titled “Failure Mechanism of pHEMT in Navigation LNA under UWB EMP” and for providing many helpful comments and suggestions, which will all prove invaluable in the revision and improvement of our paper, as well as in guiding our research in the future.

Considering the comments on the manuscript, a more detailed explanation of the numerical simulation methods and effect experiment is introduced in this paper. Besides, all the mistakes mentioned in the comments have been well-corrected. All the revised and supplementary content are marked up using the "Track Changes" function, and highlights are highlighted in yellow. We sincerely hope that the revised manuscript will help you understand our work better.

Appended to this letter is our point-to-point response to the comments raised by the reviewers. Thank you again for your invaluable comments and suggestions. I look forward to hearing from you soon in due course.

Yours sincerely,

Yonglong Li,

liylong27@mail2.sysu.edu.cn,

College of Electronics and Communication Engineering,

Sun Yat-sen University – Shenzhen Campus.

Response to Reviewer2 Comments

The paper is interesting and includes both theoretical and experimental contributions to investigate the operation of a receiver, in particular the LNA stage which is likely to be the first to be hit by an incoming EM pulse.

Reply: We really appreciate your carefulness and conscientiousness. Your suggestions are really valuable and helpful for revising and improving our paper. According to your suggestions, we have made the following revisions to this manuscript:

Point 1: My most compelling comment to the study reported in this paper is that the focus on pHEMT is not supported by clear evidence that the hardware used in the experimental verification is effectively based on a pHEMT device. Lines 229-231 describe the LNA (including a picture in Fig. 13) but there is no indication of the manufacturer or source of the LNA, leaving the reader without the opportunity to replicate the results of the paper.

Response 1: We would like to thank the reviewer for reminding us of the negligence. As mentioned in the comment, the structure and source of the pHEMT-based LNA used in the experiment are indeed not specified in detail in the previous manuscript. The circuit schematic of the LNA sample experimented has been added to give the complete structure, which is shown in Figure 13. (b) in the new manuscript. And the details of the sample have been added in the corresponding text. This design operates on a single +5V supply (VCC) and is internally matched to 50 Ohms.

Figure 13 (b) Circuit schematic of the LNA

According to the circuit schematic shown, we manufactured the LNA sample by ourselves using a pHEMT-based Low Noise MMIC amplifier(MCL 545G) produced by Mini-Circuits. As the injection of the UWB EMP, the pHEMTs in MCL-545G are significantly jammed, consequently causing temporary failure of the entire LNA. We believe that the reader could well replicate the experimental results with the revised detailed specification.

This partial revision can be viewed on lines 244-252 of the new manuscript.

Other less critical comments I would offer to the authors:

Reply: We would like to thank the reviewer for pointing out the mistakes we made, and the corresponding texts and figures have been revised according to the comments.

Point 2: Line 128 states that "the first or second derivative of Gaussian pulse is commonly used to represent UWB EMP [24,25]. Figure 3 illustrates the time and frequency domain characteristic waveforms of UWB EMP, which approximates the waveform used in this paper." The authors should specify that Fig. 3 shows the first derivative.

Response 2: Revised text: “Figure 3 illustrates the time and frequency domain characteristic waveforms of the first derivative of Gaussian pulse, which approximates the waveform of UWB EMP used in this paper.”

This partial revision can be viewed on lines 139-143 of the new manuscript.

Point 3: Fig. 5 shows the temperature variation of the device excited by a train of impulses. However, lines 143-154 do not confirm that the excitation is a train of impulse. While the explanation of the authors in those lines is logical, the type of excitation the device is subject to, is not clearly defined: is it a train of pulses? Is my interpretation correct? The authors should clarify.

Response 3: Revised text: “Based on the simulation model given above, when a train of UWB EMP with an amplitude of 8.5 V, a pulse width of 1 ns and a repetition frequency of 100 kHz is injected, the variation of peak temperature over time inside the pHEMT is shown in Figure 5.”

The excitation is indeed a train of impulse. We are sorry for not specifying. This partial revision can be viewed on lines 155-159 of the new manuscript.

Eq. (4) should be referred to as voltage gain and it should read 20*log |Vout|/|Vin|.

Revised equation (4):

This partial revision can be viewed on lines 235 of the new manuscript.

Point 4: Line 231 should report the noise figure as 1dB rather than just 1 - or provide the correct interpretation of that number.

Response 4: Revised text: “The operating frequency is 1.1-1.6 GHz, the amplification gain is 30 dB, and the noise figure is 1.0 dB.”

We are sorry for the typo error. This partial revision can be viewed on lines 247-249 of the new manuscript.

Other supplements:

About research design and methodology

We focus on the failure mechanism of pHEMT in Navigation LNA under UWB EMP by means of Numerical simulation and experimental demonstration.

Numerical simulation: Numerical simulation plays a very important role in the study of the effect mechanism of EMP in microelectronic devices. Based on the semiconductor physics equation, the transient response of pHEMT under the jamming of UWB EMP can be obtained by numerical simulation using Sentaurus-TCAD software so as to observe the whole process of device failure. The simulation results can provide a reference for failure mechanism analysis of the device, while the distribution of internal physical quantities cannot be attainable by experiments.

Experimental demonstration: The effect experiment can monitor the output performance of the device in real-time, which is a relatively intuitive method to study the law of UWB effect. The effect data obtained by experiments are more realistic and have high confidence.

The partial supplementary content could be viewed on lines 89-93 and lines 227-229.

Round 2

Reviewer 1 Report

The author has completed all revisions required.